# Orbital torque in magnetic bilayers

Dongjoon Lee[1,2,11], Dongwook Go[3,4,11], Hyeon-Jong Park[1,11], Wonmin Jeong[2,5], Hye-Won Ko[6], Deokhyun Yun[2,7], Daegeun Jo[8], Soogil Lee[9], Gyungchoon Go[6], Jung Hyun Oh[5], Kab-Jin Kim[6], Byong-Guk Park[9], Byoung-Chul Min[2], Hyun Cheol Koo[1,2], Hyun-Woo Lee[8,10 ✉], OukJae Lee[2 ✉] & Kyung-Jin Lee[6 ✉]

The orbital Hall effect describes the generation of the orbital current flowing in a perpendicular direction to an external electric field, analogous to the spin Hall effect. As the orbital current carries the angular momentum as the spin current does, injection of the orbital current into a ferromagnet can result in torque on the magnetization, which provides a way to detect the orbital Hall effect. With this motivation, we examine the current-induced spin-orbit torques in various ferromagnet/heavy metal bilayers by theory and experiment. Analysis of the magnetic torque reveals the presence of the contribution from the orbital Hall effect in the heavy metal, which competes with the contribution from the spin Hall effect. In particular, we find that the net torque in Ni/Ta bilayers is opposite in sign to the spin Hall theory prediction but instead consistent with the orbital Hall theory, which unambiguously confirms the orbital torque generated by the orbital Hall effect. Our finding opens a possibility of utilizing the orbital current for spintronic device applications, and it will invigorate researches on spin-orbit-coupled phenomena based on orbital engineering.

[1] KU-KIST Graduate School of Converging Science and Technology, Korea University, Seoul 02841, Korea. [2] Center for Spintronics, Korea Institute of Science and Technology, Seoul 02792, Korea. [3] Peter Grünberg Institut and Institute for Advanced Simulation, Forschungszentrum Jülich and JARA, 52425 Jülich, Germany. [4] Institute of Physics, Johannes Gutenberg University Mainz, 55099 Mainz, Germany. [5] Department of Materials Science and Engineering, Korea University, Seoul 02841, Korea. [6] Department of Physics, Korea Advanced Institute of Science and Technology, Daejeon 34141, Korea. [7] Department of Electrical Engineering, Korea University, Seoul 02841, Korea. [8] Department of Physics, Pohang University of Science and Technology, Pohang 37673, Korea. [9] Department of Materials Science and Engineering, Korea Advanced Institute of Science and Technology, Daejeon 34141, Korea. [10] Asia Pacific Center for Theoretical Physics, Pohang 37673, Korea. [11] These authors contributed equally: Dongjoon Lee, Dongwook Go and Hyeon-Jong Park. ✉email: hwl@postech.ac.kr; ojlee@kist.re.kr; kjlee@kaist.ac.kr

While it is often assumed that the electronic orbital degree of freedom is frozen in solids, the orbital transport, a flow of orbital angular momentum, is generally present in nonequilibrium[1]. It has been suggested that an electric field generates an orbital Hall current, i.e., orbital Hall effect (OHE)[2–4]. Analogous to the definition of spin Hall effect (SHE)[5–9], OHE refers to an excitation of an orbitally polarized current flowing along the perpendicular direction to an electric field. So far, intrinsic mechanisms of OHE have been theoretically investigated in different classes of materials, including transition metals[10], semiconductors[2], and two-dimensional materials[11–14], suggesting that OHE is much stronger than SHE by an order of magnitude in most cases. Moreover, OHE does not require spin-orbit coupling (SOC) for its emergence because an external electric field can directly interact with orbital degree of freedom. This is in clear contrast to the case of SHE, where an electric field interacts indirectly with spin degree of freedom via SOC. For this reason, while utilizing SHE inevitably involves materials with heavy elements, such a constraint on material choice may be lifted if OHE is used instead. These properties of OHE promote orbitronics that utilizes the orbital current as an information carrier to be a promising candidate for future information technology[15], in complement to spintronics.

Intrinsic OHE also brings a perspective in understanding the mechanism of intrinsic SHE. Out of competing mechanisms of SHE, the intrinsic mechanism based on Berry phase received attention[16,17], which is motivated partly by experimental reports[18,19] that the instrinsic SHE is dominant in Pt and Ta, important materials for spintronic device applications[20–23]. Nonetheless, as extrinsic mechanisms of SHE are known to exist[9,24,25], extrinsic contributions to OHE may also exist, which have not been investigated so far however. Therefore, we focus on only intrinsic OHE here. In the intrinsic mechanism, which is driven by wave function correlations without resorting to impurity scatterings, OHE is accompanied by SHE in the presence of SOC, which correlates the spin and orbital parts of the electronic wave function. In nomal metals (NMs) such as $4d$ and $5d$ heavy metals, the relative sign between orbital Hall conductivity $\sigma_{OH}^{NM}$ and spin Hall conductivity $\sigma_{SH}^{NM}$ is found to be determined by the correlation $R_{NM} = \langle \mathbf{L} \cdot \mathbf{S} \rangle_{NM}$ where $\mathbf{L}$ and $\mathbf{S}$ are orbital and spin angular momenta, respectively (i.e., $\sigma_{SH}^{NM} \sim R_{NM}\sigma_{OH}^{NM}$; Fig. 1a)[3,10]. $R_{NM}$ changes its sign from negative to positive as the outermost $d$ shell of heavy metals is progressively filled. Since the calculated $\sigma_{OH}^{NM}$ is positive for all examined $4d$ and $5d$ heavy metals, the sign of $R_{NM}$ naturally explains a Hund-rule-type behavior of the sign of $\sigma_{SH}^{NM}$ in those NMs.

In this work, we use this sign relation between the orbital Hall current and spin Hall current to experimentally confirm the existence of OHE in transition metal bilayers consisting of a NM and a ferromagnet (FM), following a theoretical prediction[26] that OHE can generate a torque (orbital torque) when an orbital Hall current is injected into FM. In particular, we highlight the importance of the orbital-to-spin conversion occuring within FM as it results in an additional contribution to spin-orbit torque (SOT) on the magnetization, which competes with the conventional contribution caused by the intrinsic SHE of NM. Recent experiments on surface-oxidized Cu/ferromagnet structures converged to an idea that the orbital angular momentum is in action for the SOT[27–29], which is generated from the interfacial orbital Rashba state[30]. Here, the spin current's contribution does not seem significant due to the negligible SOC of Cu. In general, however, both spin current and orbital current generate the torque on the magnetization, and thus it is crucial to devise an experimental protocol to unambigiuously verify the presence of the orbital current in a situation where the spin current contribution to the SOT cannot be ignored.

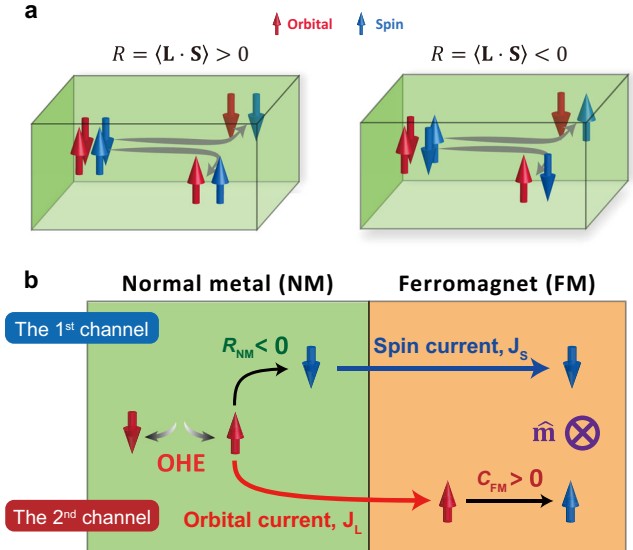

**Fig. 1 Schematic Illustration of the OHE and orbital torque. a** When an electric field is applied along the horizontal direction, a transverse orbital current is generated owing to the OHE. This orbital current is converted to a spin current through the spin-orbit coupling. Depending on the sign of spin-orbit correlation $R = \langle \mathbf{L} \cdot \mathbf{S} \rangle$, the spin polarization of the resulting spin current is either parallel (left panel) or antiparallel (right panel) to the orbital polarization of the orbital current. **b** Two channels for generating the torque in FM/NM bilayers. The first channel: An orbital current $J_L$ created through the OHE is converted to a spin current $J_S$ within the NM. For a negative spin-orbit correlation of NM ($R_{NM} < 0$), the direction of spin polarization carried by $J_S$ is the opposite to that of the orbital polarization carried by $J_L$, which is the case of Ta. This spin current is injected into a FM and exerts a torque on the magnetization $\hat{\mathbf{m}}$. The second channel: $J_L$ created through the OHE in the NM is injected into a FM in which $J_L$ is converted to $J_S$. This $J_S$ exerts a torque on $\hat{\mathbf{m}}$, which we call the orbital torque. For FMs with a positive orbital-to-spin conversion efficiency of FM $C_{FM}$ (such as Fe, Co, CoFe, and Ni), the direction of the spin polarization carried by $J_S$ is the same with that of the orbital polarization carried by $J_L$. When the second channel supplies a stronger torque than the first channel and the contributions from the two channels have the opposite signs, the sign of the net torque is the opposite to that expected for the spin Hall effect of NM.

## Results

**Detection scheme of the orbital Hall effect.** It is not simple to distinguish the orbital torque from the conventional SOT (arising from the spin Hall current injection) since the orbital torque and the SOT have identical properties with regard to all symmetry transformations such as time reversal, space inversion, and mirrors. However, the two torques depend on the spin-orbit correlations in NM and FM in different ways. Thus an observation of such difference can constitute an experimental confirmation of the OHE. Figure 1b illustrates two channels that give rise to the two torques. According to the orbital torque theory[26], OHE is the initial process for both channels: An electric field creates an orbital current $J_L$ in NM through OHE. In the first channel, the orbital current $J_L$ is converted to the spin current $J_S$ within NM through SOC of NM. This spin current is then injected into FM and produces a conventional SOT. The sign of the resulting SOT is determined by the sign of the $\sigma_{SH}^{NM}$ ($\sim R_{NM}\sigma_{OH}^{NM}$). On the other hand, the second channel corresponds to the orbital torque: The orbital current $J_L$ is injected into FM. Within the FM, this orbital current $J_L$ is then converted to the spin current $J_S$ through SOC of FM and exerts a torque, i.e., orbital torque. For the second channel, the sign of orbital torque is determined by the sign

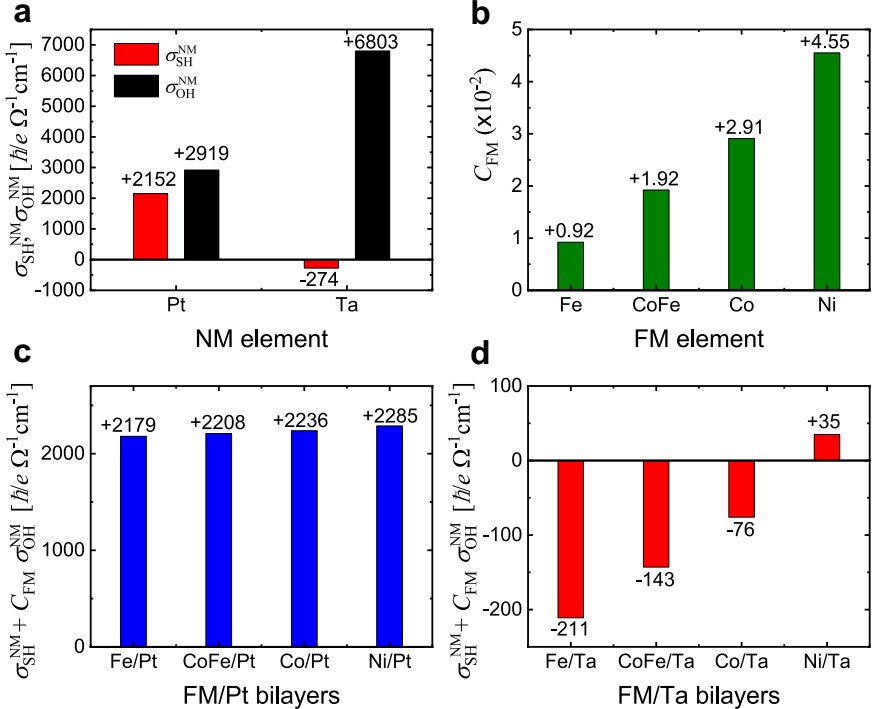

**Fig. 2 First-principles results. a** Spin Hall conductivity $\sigma_{\mathrm{SH}}^{\mathrm{NM}}$ and orbital Hall conductivity $\sigma_{\mathrm{OH}}^{\mathrm{NM}}$ of Pt and $\beta$-Ta. **b** Orbital-to-spin conversion efficiency $C_{\mathrm{FM}}$ of Fe, CoFe, Co, and Ni. **c** $\sigma_{\mathrm{SH}}^{\mathrm{NM}} + C_{\mathrm{FM}}\sigma_{\mathrm{OH}}^{\mathrm{NM}}$ of FM/Pt bilayers (FM = Fe, Co, CoFe, and Ni). **d** $\sigma_{\mathrm{SH}}^{\mathrm{NM}} + C_{\mathrm{FM}}\sigma_{\mathrm{OH}}^{\mathrm{NM}}$ of FM/Ta bilayers (FM = Fe, Co, CoFe, and Ni). In (**b**), the value of $C_{\mathrm{FM}}$ depends on the ratio $T_L/T_S$, where the spin (orbital) transparency $T_S$ ($T_L$) denotes the likelihood of spin (orbital) Hall current gets transmitted through the FM/NM interface. The ratio $T_L/T_S$ is assumed to be 0.3 in (**b**), considering that the orbital relaxation is expected to be faster than the spin relaxation.

product of the orbital Hall conductivity of the NM ($\sigma_{\mathrm{OH}}^{\mathrm{NM}}$) and the effective spin-orbit correlation $C_{\mathrm{FM}}$ in the FM. Here, $C_{\mathrm{FM}}$ describes how much spin accumulation is induced in the FM in response to an orbital Hall current injected from the NM (see section S1). The sign of $C_{\mathrm{FM}}$ is essentially the same with that of the FM's spin-orbit correlation $R_{\mathrm{FM}} = \langle \mathbf{L} \cdot \mathbf{S} \rangle_{\mathrm{FM}}$, and its magnitude is roughly proportional to $R_{\mathrm{FM}}$ that corresponds to the orbital-to-spin conversion for an internally generated orbital Hall current within the FM. Thus the net SOT of FM/NM bilayer consists of not only a spin Hall contribution proportional to $\sigma_{\mathrm{SH}}^{\mathrm{NM}}$ but also an additional orbital Hall contribution proportional to $C_{\mathrm{FM}}\sigma_{\mathrm{OH}}^{\mathrm{NM}}$.

Given these two contributions, there are three possible cases for the sign of the net SOT (conventional SOT + orbital torque): (i) When $\sigma_{\mathrm{SH}}^{\mathrm{NM}} \cdot C_{\mathrm{FM}}\sigma_{\mathrm{OH}}^{\mathrm{NM}} > 0$, both channels generate torques with the same sign so that the OHE effectively strengthens the SHE. (ii) When $\sigma_{\mathrm{SH}}^{\mathrm{NM}} \cdot C_{\mathrm{FM}}\sigma_{\mathrm{OH}}^{\mathrm{NM}} < 0$ and the spin Hall contribution is larger in magnitude than the orbital Hall one, the sign of net SOT is determined by the sign of $\sigma_{\mathrm{SH}}^{\mathrm{NM}}$ and the OHE effectively weakens the SHE. (iii) When $\sigma_{\mathrm{SH}}^{\mathrm{NM}} \cdot C_{\mathrm{FM}}\sigma_{\mathrm{OH}}^{\mathrm{NM}} < 0$ and the spin Hall contribution is smaller in magnitude than the orbital Hall one, the sign of the net SOT is opposite to that expected from the sign of $\sigma_{\mathrm{SH}}^{\mathrm{NM}}$. For the cases (i) and (ii), distinguishing the OHE from the SHE is difficult because it requires a detailed quantitative analysis of the magnitudes of each contribution, which is often ambiguous. For the case (iii), however, the sign reversal of net SOT by the OHE is unique and unambiguous. Thus if the sign reversal in the case (iii) is realized in experiments, it convincingly supports the presence of the OHE and the associated orbital torque.

To check whether or not the case (iii) can be realized in real materials, we compute $\sigma_{\mathrm{SH}}^{\mathrm{NM}}$ and $\sigma_{\mathrm{OH}}^{\mathrm{NM}}$ of NMs (Pt and Ta), and

$C_{\mathrm{FM}}$ of FMs (Fe, CoFe, Co, and Ni), based on a first-principles calculation (see section S1). Using the signs and magnitudes of calculated $\sigma_{\mathrm{SH}}^{\mathrm{NM}}$, $\sigma_{\mathrm{OH}}^{\mathrm{NM}}$, and $C_{\mathrm{FM}}$, we attempt to find out which combination of NM and FM would be most probable to realize the case (iii).

Figure 2 summarizes the calculation results. We find that for Pt, $\sigma_{\mathrm{SH}}^{\mathrm{Pt}}$ and $\sigma_{\mathrm{OH}}^{\mathrm{Pt}}$ are both positive and comparable in magnitude, whereas for Ta, $\sigma_{\mathrm{SH}}^{\mathrm{Ta}}$ and $\sigma_{\mathrm{OH}}^{\mathrm{Ta}}$ have the opposite signs and $\sigma_{\mathrm{OH}}^{\mathrm{Ta}}$ is an order of magnitude larger than $\sigma_{\mathrm{SH}}^{\mathrm{Ta}}$ (Fig. 2a). Thus the case (iii) can be more easily realized in Ta than in Pt. The significant magnitude difference between $\sigma_{\mathrm{SH}}^{\mathrm{Ta}}$ and $\sigma_{\mathrm{SH}}^{\mathrm{Pt}}$ originates from their band structure difference. For instance, it was reported[10] that Ta would have a much larger spin Hall conductivity if its band structure resembled that of Pt. The calculated $\sigma_{\mathrm{SH}}^{\mathrm{NM}}$ and $\sigma_{\mathrm{OH}}^{\mathrm{NM}}$ are consistent with previous tight-binding[3,10] and first-principles[31,32] results. We also find that $C_{\mathrm{FM}}$ is positive for all of Fe, Co, and Ni (Fig. 2b), and increases with the atomic number or the number of $3d$ electrons (thus, a Hund-rule type behavior). $C_{\mathrm{FM}}$ of CoFe is close to the average of $C_{\mathrm{FM}}$ values for Co and Fe. This FM-dependence of $C_{\mathrm{FM}}$ implies that the orbital torque should be most pronounced in Ni, which is also consistent with Ni's having the strongest spin-orbit correlation $R_{\mathrm{FM}}$ among the FM materials (see section S1).

Figures 2c and 2d respectively show $\sigma_{\mathrm{net}}^{\mathrm{FM/NM}} (= \sigma_{\mathrm{SH}}^{\mathrm{NM}} + C_{\mathrm{FM}}\sigma_{\mathrm{OH}}^{\mathrm{NM}})$ for FM/Pt and FM/Ta bilayers, calculated from the results in Fig. 2A and B. The signs of $\sigma_{\mathrm{net}}^{\mathrm{FM/NM}}$ for FM/Pt bilayers are all positive, consistent with the sign of $\sigma_{\mathrm{SH}}^{\mathrm{Pt}}$. The signs of $\sigma_{\mathrm{net}}^{\mathrm{FM/NM}}$ for Fe/Ta, Co/Ta, and CoFe/Ta bilayers are all negative, consistent with the sign of $\sigma_{\mathrm{SH}}^{\mathrm{Ta}}$. However, the sign of $\sigma_{\mathrm{net}}^{\mathrm{FM/NM}}$ for Ni/Ta bilayer is positive and thus is the opposite to that expected for the sign of $\sigma_{\mathrm{SH}}^{\mathrm{Ta}}$, because $\sigma_{\mathrm{SH}}^{\mathrm{Ta}} \cdot C_{\mathrm{Ni}}\sigma_{\mathrm{OH}}^{\mathrm{Ta}} < 0$ and $|\sigma_{\mathrm{SH}}^{\mathrm{Ta}}| < |C_{\mathrm{Ni}}\sigma_{\mathrm{OH}}^{\mathrm{Ta}}|$ for this bilayer. This calculation suggests that

the Ni/Ta bilayer is the most probable system to realize the case (iii), which motivates us to experimentally measure the sign of the net SOT for various FM/NM bilayers, including the Ni/Ta bilayer.

**Experimental test of orbital torque**. We first measure the net SOT from four types of bilayers: $Co_{40}Fe_{40}B_{20}(t_{FM})$/Pt(5), $Co_{40}Fe_{40}B_{20}(t_{FM})$/Ta(4), Ni($t_{FM}$)/Pt(5), and Ni($t_{FM}$)/Ta(4) (the numbers are in nanometers and $t_{FM}$ is the FM thickness). The resistivity of Ta is ~200 μΩ·cm, and thus it is β-phase. The FM layer is passivated with Ta(2)/MgO(2) capping layers unless specified otherwise (see Methods). For the net SOT measurement, we use the spin-torque ferromagnetic resonance (ST-FMR) technique[33–37] (see Methods), which is an established method to separately measure the damping-like torque (DLT) [$\tau_{DL} \propto \hat{m} \times (\hat{m} \times \hat{y})$] and field-like torque (FLT) [$\tau_{FL} \propto (\hat{m} \times \hat{y})$] components of the SOT, where $\hat{m}$ is the unit vector along the magnetization direction and $\hat{y}$ is the direction perpendicular to both directions of current flow ($\hat{x}$) and thickness ($\hat{z}$). From the measured results, we focus only on the sign of the DLT because it reflects the signs of the spin Hall and orbital Hall conductivities.

For the ST-FMR measurement (Fig. 3a), a radio frequency (RF) current injected into a FM/NM bilayer generates oscillating SOTs, which in turn excites the magnetization precession at resonance. This magnetization precession induces a net oscillation in the magnetoresistance. Combined with the applied RF current, the resistance oscillation induces a finite DC voltage $V_{mix}$ as a function of the applied magnetic field $H$, given as[33]

$$V_{mix}(H) = V_S \frac{\triangle H^2}{(H - H_{res})^2 + (\triangle H)^2} + V_A \frac{(H - H_{res})\triangle H}{(H - H_{res})^2 + (\triangle H)^2},$$
(1)

where $V_S(V_A)$ is the symmetric (antisymmetric) term of resonance amplitude, $H_{res}$ is the resonance field, and $\triangle H$ is the half linewidth at the half maximum. The $V_S$ corresponds to the DLT, whereas the $V_A$ corresponds to the sum of the FLT and current-induced Oersted field torque.

Figure 3b–e show representative ST-FMR data of four bilayers ($t_{FM} = 5$ nm). The CoFeB/Pt bilayer (Fig. 3b) shows a negative $V_S$ corresponding to a positive spin Hall angle that has the same sign with $\sigma_{SH}^{Pt}$, whereas the CoFeB/Ta bilayer (Fig. 3c) shows a positive $V_S$ corresponding to a negative spin Hall angle that has the same sign as $\sigma_{SH}^{Ta}$. The Ni/Pt bilayer (Fig. 3d) shows a negative $V_S$, same as for the CoFeB/Pt bilayer, which is consistent with the sign of $\sigma_{SH}^{Pt}$. However, we observe the abnormal case for the Ni/Ta bilayer: this bilayer shows a negative $V_S$ (Fig. 3e), which is the opposite to that expected from the sign of $\sigma_{SH}^{Ta}$. As the sign of $V_S$ depends not only on the sign of net spin Hall conductivity $\sigma_{net}^{FM/NM}$ but also on the sign of anisotropic magnetoresistance (AMR), we check the sign of AMR of Ni/Pt and Ni/Ta bilayers. We find that they are of the same sign so that the sign of DLT for Ni/Ta bilayer is abnormal.

The abnormal DLT sign of Ni/Ta bilayer is further confirmed by the FM thickness ($t_{FM}$) dependence of $V_{mix}$ (see section S2), which gives $t_{FM}$-independent DLT and FLT efficiencies. We perform ST-FMR measurements for various $t_{FM}$ and obtain the quantity $\xi_{FMR}$, defined from $V_S/V_A$[34,35];

$$\xi_{FMR} = \frac{V_S}{V_A} \left(\frac{e}{\hbar}\right) 4\pi M_s t_{FM} t_{NM} \sqrt{1 + \frac{4\pi M_{eff}}{H_{res}}},$$
(2)

where $M_s$ is the saturation magnetization of FM, $t_{NM}$ is the thickness of NM (Pt or Ta), and $4\pi M_{eff}$ is the out-of-plane demagnetization field (see Methods). The measurement of $\xi_{FMR}$ as a function of $t_{FM}$ allows us to separately estimate the damping-like ($\xi_{DL}$) and field-like ($\xi_{FL}$) torque efficiencies, using[34,35,38]

(see Methods)

$$\frac{1}{\xi_{FMR}} = \frac{1}{\xi_{DL}} \left(1 + \frac{\hbar}{e} \frac{\xi_{FL}}{4\pi M_s t_{FM} t_{NM}}\right).$$
(3)

Equation (3) shows that the intercept of $1/\xi_{FMR}$ in the limit of $1/t_{FM} \to 0$ gives the sign of $\xi_{DL}$. As shown in Fig. 3f (CoFeB/Pt and CoFeB/Ta) and 3g (Ni/Pt and Ni/Ta) (see section S2), the signs of the intercepts, indicated by arrows, are all consistent with those of the symmetric peaks in Fig. 3b–e. Thus, the abnormal sign of the DLT for the Ni/Ta bilayer is not specific to a sample but general. As independent tests, we also experimentally examine the sign of the DLT by measuring the linewidth (or effective damping) modulation by a DC current (see section S3), and by utilizing the 2nd harmonic Hall measurement for the in-plane magnetized systems (see section S4). We obtain a consistent result with ST-FMR. Therefore, it is clear that the DLT sign of the Ni/Ta bilayer is opposite to that expected from the sign of $\sigma_{SH}^{Ta}$. We note that this abnormal sign is consistent with the above-explained case (iii), i.e., the orbital torque. It is also found that the measured Landé g-factors for Ni/Pt, Ni/Ta, CoFeB/Pt and CoFeB/Ta are consistent with the OHE scenario (section S5). Figure 3h (3i) summarizes the effective spin-Hall angle (spin Hall conductivity) for various bilayers (FM = FeB, CoFeB, Co, Ni, NM = Pt, Ta; see section S6 for further details). Note that the material-dependent variation of the effective spin Hall conductivity is in qualitative agreement with the theoretical result in Figs. 2c and 2d, which supports the OHE theory.

**Other possible mechanisms of the abnormal DLT sign**. In the previous section, our analysis focuses on the competition between the SHE and OHE for the current-induced torque, but neglects other mechanisms. To examine the effects of different mechanisms, we perform control experiments.

In addition to the bulk SHE of NM, recent studies suggested an important role of the interfacial SOC in the SOT[39–41]. As the sign of the spin polarization carried by interface-generated spin currents can be different from that of the bulk SHE of NM, we investigate the interfacial SOC effect at both interfaces of Ni/Ta bilayer: Ni/Ta interface and MgO/Ni interface. To check if the Ni/Ta interface is crucial, we carry out ST-FMR measurements for Ni(7)/Cu($t_{Cu}$)/Ta(4) samples where a Cu layer is inserted between Ni and Ta layers. We find that the DLT still has the same abnormal sign (Fig. 4a, b; see section S2). The sign of DLT is maintained regardless of $t_{Ni}$ for Ni($t_{Ni}$)/Cu(1)/Ta(4) layers (see section S2). These results imply that the interfacial SOC effect[42] from the Ni/Ta interface is not responsible for the abnormal sign. The MgO/Ni interface may also affect the sign of the DLT because the oxide layer is known to enhance the Rashba effect at the interface[43,44]. To check this possibility, we replace Ta(2)/MgO(2) capping layers with $HfO_x$(3). In the resulting $HfO_x$/Ni interface, the interaction between Ni and oxygen is expected to be strongly suppressed since the enthalpy of formation ($\approx -1120$ kJ/mol) of $HfO_x$ is much larger in magnitude than that of MgO ($\approx -600$ kJ/mol)[45]. We find that the sign of the DLT does not change with this replacement (Fig. 4c, d; see section S2), implying that the MgO/Ni interface is not crucial either. These control experiments allow us to exclude the possibility that the abnormal sign is caused by the interfacial effect.

Another possible contribution can be driven by a self-induced spin accumulation of the FM, which is known as the anomalous torque[46–48]. A recent first-principles calculation suggests that the anomalous torque can be substantial in Ni due to its pronounced spin-orbit correlation, which also contributes to the abnormal sign of the SOT due to a positive sign of $\sigma_{SH}^{Ni}$[49]. To estimate such contribution, we perform ST-FMR measurements for

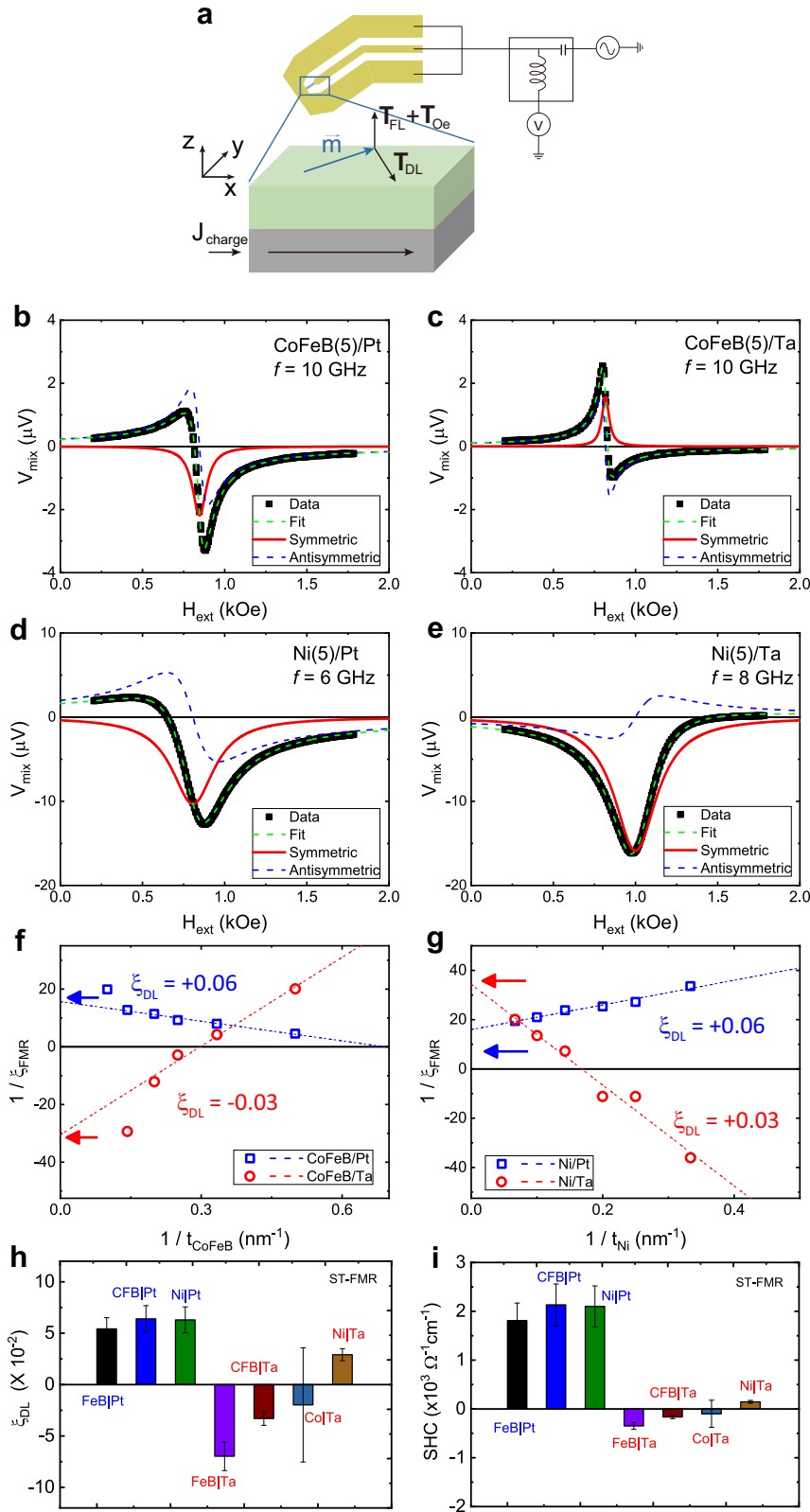

**Fig. 3 ST-FMR results of FM/NM bilayers (FM = CoFeB, Ni, NM = Pt, Ta). a** A schematic illustration of ST-FMR measurement. **b–e** $V_{mix}(H)$ of CoFeB(5)/Pt (**b**), CoFeB(5)/Ta (**c**), Ni(5)/Pt (**d**), and Ni(5)/Ta (**e**) bilayers. Symbols are experimental results and lines are fitting results with Eq. (1). **f, g** $1/\xi_{FMR}$ as a function of **f** $1/t_{FM}$ of CoFeB($t_{CoFeB}$)/NM (**f**) and Ni($t_{Ni}$)/NM (NM = Pt, Ta) (**g**). **h** The damping-like torque efficiency $\xi_{DL}$ for various bilayers (FM = FeB, CoFeB, Co, Ni, NM = Pt, Ta). **i** The effective spin Hall conductivity ($\sigma_{DL}$) for various bilayers. The error bars in (**h**) and (**i**) indicate single-standard-deviation uncertainties from the fitting.

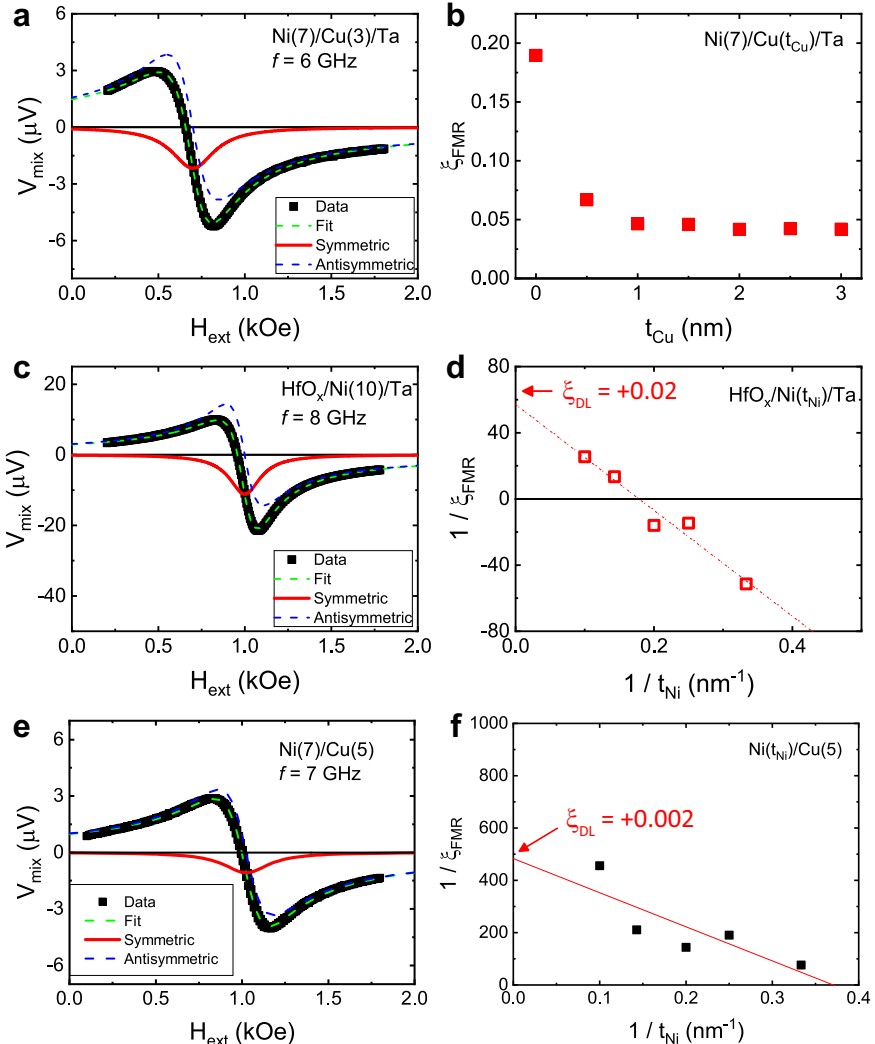

**Fig. 4 Control experiments for other possible mechanisms. a** ST-FMR spectra of a Ni(7)/Cu(3)/Ta sample. **b** $\xi_{FMR}$ as a function of the Cu-layer thickness $t_{Cu}$ for Ni(7)/Cu($t_{Cu}$)/Ta samples. **c** ST-FMR spectra of a HfO$_x$(3)/Ni(10)/Ta sample. **d** $1/\xi_{FMR}$ as a function of $1/t_{Ni}$ of HfO$_x$(3)/Ni($t_{Ni}$)/Ta samples. **e** ST-FMR spectra of a Ni(7)/Cu(5) sample. **f** $1/\xi_{FMR}$ as a function of $1/t_{Ni}$ of Ni($t_{Ni}$)/Cu samples.

Ni($t_{Ni}$)/Cu(5) bilayers (Fig. 4e, f). Since both OHE and SHE are negligible in pure Cu[50], the SOT in Ni/Cu represents a contribution from the anomalous torque. While the signs are still abnormal as expected, the magnitudes of the SOT are an order of magnitude smaller than that in Ni/Ta. Combining all these results with those shown in Figs. 3 and 4, we conclude that the abnormal sign of DLT for the Ni/Ta layer originates from the orbital torque, which in turn proves the existence of the OHE.

## Discussion

The orbital degree of freedom is a recurring theme in condensed matter physics. It is now well established that the giant tunneling magnetoresistance in MgO-based magnetic tunnel junction[51,52] is due to the orbital-dependent tunnelling[53,54]. Moreover, it is known that the theoretical limit of magnetic anisotropy[55] and the large Rashba spin splitting[56,57] can be achieved by utilizing the orbital degree of freedom. In this respect, our experimental confirmation of the orbital torque and the OHE has a number of important implications. Whereas previous attempts to strengthen the SHE are based on a material search for NM materials with strong SOC, the relation $\sigma_{SH}^{NM} \propto \sigma_{OH}^{NM} R_{NM}$ suggests that large $\sigma_{SH}^{NM}$ may be achieved alternatively by strengthening $\sigma_{OH}^{NM}$ instead. Theoretical studies of single-element materials report large $\sigma_{OH}^{NM}$

of the order of $10^4 \left(\hbar/e\right)(\Omega \cdot \text{cm})^{-1}$ for various materials[3,4,10] but this value may be enhanced further, for instance, in multi-element materials by optimizing the orbital degree of freedom. For device applications based on the current-induced torque, another interesting direction is to pursue strengthening the orbital torque by optimizing the FM material choice, since the orbital torque can be strengthened in FM with strong SOC[4]. Considering that the weak SOC material Ni can already overcome the strong SHE of Ta and reverse the net SOT sign in a Ni/Ta bilayer, this is a promising direction to pursue enhancing the torque. It thus provides a new opportunity that was not explored by present trends to optimize the NM material choice. By the way, our experiment intentionally uses well-characterized 3$d$ FMs with weak SOC to avoid possible complications by strong SOC in FMs since our work is aimed to confirm the OHE unambiguously instead of maximizing the orbital torque. To maximize the total torque, it is actually desired to choose NM and FM materials so that the orbital torque and the conventional SOT have the same sign to add them up instead of canceling each other. We expect that the orbital current confirmed in this experiment opens a new avenue of exciting opportunities for more advanced spintronic and possibly orbitronic devices. Moreover, our result suggests that the orbital engineering is an efficient pathway to enhance the spin

current that is an essential ingredient not only in spintronics but also in various branches of condensed matter physics.

## Methods

**Thin film growth and characterization**. The multilayer films were prepared on thermally oxidized Si substrates by DC/RF magnetron sputtering at room temperature. The multilayers consist of capping layer/ferromagnetic-metal (FM)/non-magnetic-metal (NM)/substrate. The combinations of the FM and NM were $Co_{40}Fe_{40}B_{20}(t_{CoFeB})/Pt(5)$, $Co_{40}Fe_{40}B_{20}(t_{CoFeB})/Ta(4)$, $Ni(t_{Ni})/Pt(5)$, and $Ni(t_{Ni})/Ta(4)$ (nominal thickness in nm). The $t_{CoFeB}$ was varied from 2 to 10 nm and $t_{Ni}$ from 3 to 15 nm. The capping layer was either $Ta(2)/MgO(2)$ or $HfO_x(3)$ to protect its underlayers. The base pressure of the chamber was maintained less than $5 \times 10^{-8}$ Torr, and the deposition rates were kept lower than 0.5 Å/s. A vibrating sample magnetometer (VSM) was used to measure the saturation magnetization ($M_s$).

**ST-FMR Device fabrication and measurement**. To prepare devices for the ST-FMR measurement, we used optical lithography and ion-milling to pattern the multilayer films into rectangular strips with 15 μm-width ($w$) and 50 μm-length ($l$). In a subsequent process, a waveguide contact made of Au (100 nm)/Ti (10 nm) was defined on top of the samples to apply a microwave current to the devices. The samples were not exposed to high temperature (>120 °C) during the fabrication process as no post-annealing was carried out. All the measurements were performed at room temperature. For ST-FMR measurement, a pulsed microwave signal in the range 4–14 GHz with a nominal output power of 10–20 dBm was applied to the samples. An in-plane external magnetic field (from −1.8 kOe to +1.8 kOe) was swept at an angle of 45°. The $V_{mix}$ was simultaneously detected with a lock-in amplifier connected to the DC port of the bias tee. By fitting $V_{mix}(H)$ with Eq. (1), we obtained $H_{res}$, $\Delta H$, $V_S$, and $V_A$ at different frequencies for each sample. The center frequency of the resonance peak ($f$) follows the Kittel equation,

$$f = (\gamma/2\pi)\sqrt{H_{res}(H_{res} + 4\pi M_{eff})},$$ where $\gamma$ is the gyromagnetic ratio and $4\pi M_{eff}$ was extracted from a fit to the Kittel equation.

**ST-FMR data analysis**. In Eq. (1), $V_S$ is proportional to the DLT ($\tau_{DL}$), whereas $V_A$ originates from the sum of FLT ($\tau_{FL}$) and Oersted field torque ($\tau_{Oe}$);

$$V_S \propto \gamma\tau_{DL} = \gamma\frac{\hbar}{2e}\frac{J_{NM}}{4\pi M_s t_{FM}}\xi_{DL}, \quad (4)$$

$$V_A \propto \gamma(\tau_{Oe} + \tau_{FL})\sqrt{1 + \frac{4\pi M_{eff}}{H_{res}}} \approx \gamma\left(\frac{t_{NM}J_{NM}}{2} + \frac{\hbar}{2e}\frac{J_{NM}}{4\pi M_s t_{FM}}\xi_{FL}\right)\sqrt{1 + \frac{4\pi M_{eff}}{H_{res}}}, \quad (5)$$

where $\hbar$ is the Planck's constant, $e$ is the electron charge, $J_{NM}$ is the current density through the NM, and $t_{NM}$ is the thickness of NM. The measured voltages are interpreted to magnetic precession angles of less than 0.5°, corresponding to a linear-response regime. Thus we ignored the contribution in $Vs$ from spin-pumping and inverse SHE[36,58], which are estimated to be less than 0.2 μV. We also measured the magnetic dead layer thickness from $M_S$ versus $t_{FM}$, but found that it does not affect the sign and even magnitude of the DLT. The $V_{mix}$ usually includes a field ($H$)-independent offset, which was eliminated in all data.

## Data availability

Data that support the findings of this study are deposited in Zenodo with the access link https://doi.org/10.5281/zenodo.5676491.

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

## Acknowledgements
We acknowledge M. D. Stiles for discussion. D.G. acknowledges discussions with Yuriy Mokrousov, Jan-Philipp Hanke, and Frank Freimuth. We gratefully acknowledge the Jülich Supercomputing Center for providing computational resources under project jiff40. D.L., H.J.P., H.W.K., G.G., J.H.O. and K-J.L. were supported by the National Research Foundation (NRF) of Korea (NRF-2020R1A2C3013302 & 2020M3F3A2A01082591 & 2015M3D1A1070465) and Samsung Electronics Co., Ltd. (Grant No. IO201019-07699-01). D.L., H.J.P., and H.W.K. were supported by the KU-KIST school project. D.L., W.J., D.Y., B.C.M., H.C.K., and O.L. were supported by the KIST Institutional Program, Institute of Information & communications Technology Planning & Evaluation (IITP) grant funded by the Korea government (MSIT) (No. 2019-0-00296) and the NRF program (NRF-2020M3F3A2A01081635). D.G., D.J., and H.-W.L. were supported by the Samsung Science and Technology Foundation (BA-1501-07 & BA-1501-51). D.G. was supported by the Deutsche Forschungsgemeinschaft (DFG, German Research Foundation)—TRR 173/2—268565370 Spin-X (project A11).

## Author contributions
D.L., D.G. and H.J.P. equally contributed to this work. H.W.L., O.L. and K.J.L. supervised the study. D.L. and W.J. prepared films/devices and performed ST-FMR measurements with help from D.Y., B.C.M., H.C.K. and O.L. D.G., H.J.P., H.W.K. and D.J. carried out theoretical calculations with help from G.G., J.H.O., H.W.L., and K.J.L. D.L., O.L and K.J.L. analyzed the results with help from H.W.L., S.L., B.G.P. and K.J.K. K.J.L., O.L., H.W.L. and H.C.K. wrote the paper. All authors discussed the results and commented on the paper.

## Competing interests
The authors declare that they have no competing interests.
