## [Peer Review File · Nature Communications]

Reviewers' Comments:

Reviewer #1:

Remarks to the Author:

In the manuscript entitled "Orbital torque in magnetic bilayers", Dongjoon Lee and collaborators search for clear signatures of the generation of orbital torque magnetic/non-magnetic metallic bilayers.

In the original theoretical work of D. Go and collaborators, the orbital torque is generated by the orbital Hall effect when an orbital Hall current, generated in a non-magnetic layer is injected into a ferromagnet. Besides the possible technological implications of a new mechanism for magnetic layer switching/manipulation, an unambiguous signature of the orbital torque is also a natural way of verifying the existence of OHE (although it is not the only feasible proposal). The authors use theoretical calculations to select the best combination of materials that allows them to determine from ST-FMR measurement that the orbital torque is excited and measured in the experimental setup.

The manipulation of the orbital degrees of freedom in quantum materials is gaining momentum, specially in orbital magnetism, with the different orbital magnetic orders presented in twisted bilayers. But also in novel 2D materials, where the orbitals play an important role in their physical properties. In this sense, I believe the present manuscript is timely and in the scope of Nature Communications.

That being said, it still needs some work before it can be accepted to publication. In general terms, although the results are solid and well presented, the introduction is confusing and lacks precision. There are some bold statements that are just not true and need to be addressed and corrected. Also, the introduction needs to be more pedagogical for the reader of Nature communication that is not familiar with orbital Hall effect. To free some space, parts of the discussion on the spin Hall effect could be easily suppressed.

Below, I list some specific points:

1) abstract "Theories suggested that an electric-field-induced generation of orbital currents, called orbital Hall effect.." This is too vague for an abstract. What theories? it can at least contain a reference.

2) Also " OHE is fundamental process, and spin currents are subsequently converted from orbital currents" This statement is not necessary for the conclusions of the work. I do not see the need to state, along the manuscript, that the OHE is the fundamental process. It might be in some situations and not in others. Unless the authors are absolutely sure and can provide strong indications that this is always the case, I believe this type of statement is unnecessary.

3) " Theories to understand the SHE are directed toward the intrinsic SHE, which does not resort to spin-dependent scattering caused by disorder. " This is plain wrong. There are several theoretical works devoted to the understanding of the SHE thru spin-depending scattering caused by disorder. Different mechanisms are discussed. It is an extensive literature. Personally, I believe the first 2-3 paragraphs have to be re-written. Similarly to this statement there are several others about the lack of understanding of the SHE that are not precise. What I do agree is that the works on OHE bring the PERSPECTIVE of analysing the SHE in terms of the OHE, which is different from what is written.

4) The authors use theoretical calculations to select the best pair of materials for the orbital torque and one of their choices is Ta, which has $OHE \gg SHE$. Can the authors provide a physical insight on why? A small sentence of the causes could be appreciated by the reader.

5) The confirmation of the OHE in this experiment is rather indirect. Is there any way to measure the OHE directly?

6) In the SM "the Hund's rule provides an intuitive explanation for this sign difference, which is

verified by the extensive tight-binding calculation (Ref. 10) " Ref 10 of SM does not exist.

Reviewer #2:

Remarks to the Author:

This manuscript theoretically discusses and experimentally demonstrates a new mechanism of a current-induced torque, coined orbital torque, in ferromagnet/nonmagnet (FM/NM) heterostructures, previously overlooked to a great extent. The orbital torque stems from an orbital angular momentum flow in solids in response to an applied electric field. The claim, summarized in the Introduction and supported by the first principle calculations in the SI, is that this orbital angular momentum flow can be converted into a spin current once absorbed by a ferromagnetic material, hence exert a spin torque on the magnetization. The authors conduct an extensive experimental investigation in various NM and FM combinations and use different experimental techniques/approaches to test this hypothesis. They achieve the unambiguous conclusion that the orbital torque can be strong and act in concert or against the spin Hall torque depending on the choice of the NM/FM system.

Since the discovery of spin-orbit torques, there is an ongoing quest to understand the underlying physics and maximize the torque/current ratio in magnetic heterostructures. The present manuscript is a crucial step in this quest, potentially advancing our understanding of the spin-orbit torques in archetypal spintronic systems and providing the means to tune them by materials engineering. The manuscript is written pedagogically with an accessible language and has a good balance between theory and experiment. I believe that it will attract widespread attention from the spintronics community and beyond. Overall, I would support the publication of this work in Nature Communications if the authors can clarify the issues listed below.

- The sign of V_s in the ST-FMR measurement depends on the sign of the anisotropic magnetoresistance. Since Ni is not a common material used in spintronic devices, and the AMR of ultrathin films can significantly deviate from their bulk behavior, a standard AMR characterization in the Ni-based systems would be useful to confirm that the 'abnormal' torque behavior is not an experimental artifact.
- The interfacial quality of Ni-based heterostructures should be adequately characterized and discussed to eliminate the possibility of new torques induced by strong interfacial alloying. Depending on the outcome, this possibility should be discussed in the relevant section of the manuscript.
- The resonance peaks of the Ni-based structures are significantly broader with respect to the CoFeB-based structures. Why?
- Can the authors comment on the implication of these findings for other spintronic systems (magnetic insulator/NM, AFM/NM, etc.) and effects (SMR, SSE, etc.)?
- The control experiment for the self-induced anomalous torque should be performed on the Ni layer alone and not on the Ni/Cu bilayer. Since Cu will have a significant shunting effect, it could be a good comparison to Ni/Pt but not to Ni/Ta, in which the authors observe the 'abnormal' behavior. An additional test could be Ni/Ti, provided that Ti is not a source of orbital angular momentum.

Dear Reviewers,

We sincerely appreciate your valuable time on evaluations and comments that have helped improve our manuscript. Please find below our point-by-point responses to all comments. The corresponding modifications are incorporated in the revised manuscript (marked in **blue**). We believe that the revised manuscript can now be published in *Nature Communications*.

Yours sincerely,

OukJae Lee, Hyun-Woo Lee, Kyung-Jin Lee on behalf of all co-authors

Reviewer #1 (Remarks to the Author):

In the manuscript entitled "Orbital torque in magnetic bilayers", Dongjoon Lee and collaborators search for **clear signatures** of the generation of orbital torque magnetic/non-magnetic metallic bilayers.

In the original theoretical work of D. Go and collaborators, the orbital torque is generated by the orbital Hall effect when an orbital Hall current, generated in a non-magnetic layer is injected into a ferromagnet. Besides the possible technological implications of a new mechanism for magnetic layer switching/manipulation, **an unambiguous signature of the orbital torque is also a natural way of verifying the existence of OHE** (although it is not the only feasible proposal). The authors use theoretical calculations to select the best combination of materials that allows them to determine from ST-FMR measurement that the orbital torque is excited and measured in the experimental setup.

The manipulation of the orbital degrees of freedom in quantum materials is gaining momentum, specially in orbital magnetism, with the different orbital magnetic orders presented in twisted bilayers. But also in novel 2D materials, where the orbitals play an important role in their physical properties. In this sense, **I believe the present manuscript is timely and in the scope of Nature Communications.**

→ We thank the reviewer for the positive evaluation of our manuscript.

That being said, it still needs some work before it can be accepted to publication. In general terms, although the results are solid and well presented, the introduction is confusing and lacks precision. There are some bold statements that are just not true and need to be addressed and corrected. Also, the introduction needs to be more pedagogical for the reader of Nature communication that is not familiar with orbital Hall effect. To free some space, parts of the discussion on the spin Hall effect could be easily suppressed.

→ We thank the reviewer for reading our manuscript carefully and suggesting how to improve the manuscript's readability. The intro part has been extensively revised as recommended. We also note that some of our authors have a recent preprint [arXiv: 2107.08478] discussing various possible approaches to quantify OHE phenomena. The

potential experiments are outlined in the response #5. We hope the update satisfies the reviewer.

Below, I list some specific points:

1) abstract "Theories suggested that an electric-field-induced generation of orbital currents, called orbital Hall effect.." This is too vague for an abstract. What theories? it can at least contain a reference.

→ We thank the reviewer for pointing out this ambiguity. The abstract is completely changed to:

“The orbital Hall effect describes the generation of the orbital current flowing in a perpendicular direction to an external electric field, analogous to the spin Hall effect. As the orbital current carries the angular momentum as the spin current does, injection of the orbital current into a ferromagnet can result in torque on the magnetization, which provides a way to detect the orbital Hall effect. With this motivation, we examine the current-induced spin-orbit torques in various ferromagnet/heavy metal bilayers by theory and experiment. Analysis of the magnetic torque reveals the presence of the contribution from the orbital Hall effect in the heavy metal, which competes with the contribution from the spin Hall effect. In particular, we find that the net torque in Ni/Ta bilayers is opposite in sign to the spin Hall theory prediction but instead consistent with the orbital Hall theory, which unambiguously confirms the orbital torque generated by the orbital Hall effect. Our finding opens a possibility of utilizing the orbital current for spintronic device applications, and it will invigorate researches on spin-orbit-coupled phenomena based on orbital engineering.”

For the suggestion of the reference, please note that the Nature Communications journal style does not recommend adding references to the abstract [“an abstract of approximately 150 words, unreferenced” on <https://www.nature.com/documents/ncomms-submission-guide.pdf>]. Instead, the 1st paragraph of INTRODUCTION contains details about the theories and related references [1-4,10-15].

2) Also " OHE is fundamental process, and spin currents are subsequently converted from orbital currents" This statement is not necessary for the conclusions of the work. I do not see the need to state, along the manuscript, that the OHE is the fundamental process. It might be in some situations and not in others. Unless the authors are absolutely sure and can provide strong indications that this is always the case, I believe this type of statement is unnecessary.

→ We agree with this comment. In the revised manuscript (INTRODUCTION, 2nd paragraph), we deleted the above sentence quoted by the reviewer. We also removed expressions that give an impression that the OHE is the fundamental process. We kept only explanations about the sign relation between the OHE and SHE, which is essential for our work.

3) " Theories to understand the SHE are directed toward the intrinsic SHE, which does not resort to spin-dependent scattering caused by disorder. " This is plain wrong. There are several theoretical works devoted to the understanding of the SHE thru spin-depending scattering caused by disorder. Different mechanisms are discussed. It is an extensive literature. Personally, I believe the first 2-3 paragraphs have to be re-written. Similarly to this statement there are several others about the lack of understanding of the SHE that are not precise. What I do agree is that the works on OHE bring the PERSPECTIVE of analysing the SHE in terms of the OHE, which is different from what is written.

→ We realized that our statement (quoted by the reviewer above) does not correctly represent the current status of the ongoing research works on the SHE. To remedy this problem, we made it clear in the revised manuscript that we focus on the only intrinsic mechanisms of SHE and OHE. The revised parts are as follows:

“Intrinsic OHE also brings a perspective in understanding the mechanism of intrinsic SHE. Out of competing mechanisms of SHE, the intrinsic mechanism based on Berry phase received attention^{16,17}, which is motivated partly by experimental reports^{18,19} that the intrinsic SHE is dominant in Pt and Ta, important materials for spintronic device applications²⁰⁻²³. Nonetheless, as extrinsic mechanisms of SHE are known to exist^{9,24,25}, extrinsic contributions to OHE may also exist, which have not been investigated so far however. Therefore, we focus on only intrinsic OHE here. In the intrinsic mechanism, which is driven by wave function correlations without resorting to impurity scatterings, OHE is accompanied by SHE in the presence of SOC, which correlates the spin and orbital parts of the electronic wave function.”

4) The authors use theoretical calculations to select the best pair of materials for the orbital torque and one of their choices is Ta, which has OHE >>SHE. Can the authors provide a physical insight on why? A small sentence of the causes could be appreciated by the reader.

→ According to first-principles calculations (Fig. 2A of the main text), the orbital Hall conductivity (OHC) of Ta is about twice larger than the OHC of Pt, whereas the spin Hall conductivity (SHC) of Ta is about eight times smaller in magnitude than the SHC of Pt. Therefore, $OHC \gg SHC$ of Ta is caused not by its large OHC but by its small SHC. In this respect, we below discuss about the small SHC of Ta.

β -Ta has a much smaller SHC in magnitude than fcc Pt, partly because β -Ta has a weaker atomic spin-orbit coupling than fcc Pt. But this difference is insignificant. The more critical difference originates from the band structure in the absence of the spin-orbit coupling since it determines how effective the spin-orbit coupling can be. The spin-orbit coupling is weaker than other energies such as the crystal field and the electron hopping. Thus, the spin-orbit coupling is usually overwhelmed by other energies. It can play significant roles only when the band structure in the absence of the spin-orbit coupling develops near-degeneracies near the Fermi energy. In this respect, fcc Pt is quite optimal, but β -Ta is far from optimal. For instance, Tanaka *et al.* [discussion about Fig. 5(a) in Ref. 10] reported that the SHC of Ta would be about an order of magnitude larger if Ta shared a similar band structure as Pt. To explain this point, we added the following sentence to the manuscript: “The significant magnitude difference between σ_{SH}^{Ta} and σ_{SH}^{Pt} originates from their band structure difference. For instance, it was reported [10] that Ta would have a much larger spin Hall conductivity if its band structure resembled that of Pt.”

5) The confirmation of the OHE in this experiment is rather indirect. Is there any way to measure the OHE directly?

→ This is an excellent but also tough question. Measuring the accumulation of the orbital angular momentum (OAM) at the boundary of the sample is probably the most direct way to confirm the OHE experimentally. However, it is a challenging measurement because, in principle, the spin angular momentum (SAM) accumulation is always accompanied by the OAM accumulation and it is not easy to distinguish the OAM from the SAM. Notably, some of our authors have a recent preprint [arXiv:2107.08478] that discusses possible experimental approaches to quantify the OHE phenomena. Here, we summarize them.

(a) One can approach this task by using the magneto-optical Kerr effect (MOKE) [Science 306, 1910 (2004), PRL 119, 087203 (2017)], which has been previously employed to detect the SAM accumulation (i.e., the direct measurement of spin-Hall effect). In principle, this is

an ideal experimental tool to probe the OAM since photons dominantly interact with the orbital part of the wave function. However, the SAM also affects the measured signal since the SAM couples to the OAM through the SOC. Thus it is desired to develop a method to differentiate the OAM contribution to the MOKE signal from the corresponding SAM contribution, which can be highly challenging for strong SOC materials such as Pt and Ta. Probably, this approach is more suited for the OAM accumulation detection in light elements with weak SOC.

(b) The X-ray magnetic circular dichroism (XMCD) can separately quantify the SAM and the OAM [Coord. Chem. Rev. 277-278, 95 (2014)]. However, this technique has a spatial resolution problem in the thickness direction since X-rays easily penetrate thin films and thus probe simultaneously both the top and bottom surfaces, where the OAM accumulations have opposite signs. Hence it is desired to devise a method to overcome this resolution problem.

(c) In principle, any light probe that interacts with the magnetic moment can be used to detect the OHE. For example, vortex beams [Phys. Rev. A. 45, 8185 (1992), Phys. Rev. Lett. 122, 237401 (2019)] carry finite photon OAM in addition to the photon spin so that they are expected to interact with the OAM in materials efficiently.

(d) Still another approach is to carry out the torque measurements with a systematically prepared series of FM / NM structures. The orbital torque (OT) by the orbital current injection into the FM is similar to the spin torque (ST) by the spin-current injection, but it is mediated by the SOC of the FM rather than that of the NM since the OAM does not directly interact with the local magnetic moment. Therefore, one can investigate the OHE through the torque measurement for a series of heterostructures with FMs having strong SOC but NMs having weak SOC. If this measurement shows a systematic variation of the torque with the SOC strength in the FM, it is strong evidence of the OHE.

6) In the SM "the Hund's rule provides an intuitive explanation for this sign difference, which is verified by the extensive tight-binding calculation (Ref. 10) " Ref 10 of SM does not exist.

➔ Reference number (10) refers to Ref. #10 [Tanaka et al. Phys. Rev. B. 77, 165117 (2008)] in the main manuscript. For convenience, the same paper is added to the reference list of SM. Thank you for pointing it out.

Reviewer #2 (Remarks to the Author):

This manuscript theoretically discusses and experimentally demonstrates a new mechanism of a current-induced torque, coined orbital torque, in ferromagnet/nonmagnet (FM/NM) heterostructures, previously overlooked to a great extent. The orbital torque stems from an orbital angular momentum flow in solids in response to an applied electric field. The claim, summarized in the Introduction and supported by the first principle calculations in the SI, is that this orbital angular momentum flow can be converted into a spin current once absorbed by a ferromagnetic material, hence exert a spin torque on the magnetization. The authors conduct an **extensive experimental investigation** in various NM and FM combinations and use different experimental techniques/approaches to test this hypothesis. **They achieve the unambiguous conclusion that** the orbital torque can be strong and act in concert or against the spin Hall torque depending on the choice of the NM/FM system.

Since the discovery of spin-orbit torques, there is an ongoing quest to understand the underlying physics and maximize the torque/current ratio in magnetic heterostructures. **The present manuscript is a crucial step in this quest, potentially advancing our understanding of the spin-orbit torques** in archetypal spintronic systems and providing the means to tune them by materials engineering. The manuscript is written pedagogically with an accessible language and has a good balance between theory and experiment. I believe that it will attract widespread attention from the spintronics community and beyond. Overall, **I would support the publication of this work in Nature Communications if the authors can clarify the issues listed below.**

→ We thank the reviewer for the positive evaluation of our manuscript. We have provided below our responses to all comments by the reviewer. In particular, we present the AMR results for Ni/Ta and Ni/Pt to rule out the possibility of AMR abnormality (see the 1st response). We performed ST-FMR measurement on MgO/Ni/substrate to measure the spin Hall angle (SHA) of Ni single layer. We also estimated the spin Hall conductivity (SHC) of Ni from experiments for Ni/Cu bilayers with considering the shunting effect and assuming that the SHC of Ni/Cu is entirely caused by the SHE of Ni. The results show that the SHE of Ni (or anomalous torque contribution) cannot explain the reversed sign in Ni/Ta (see the last response). We hope our responses satisfy the reviewer.

- The sign of V_s in the ST-FMR measurement depends on the sign of the anisotropic magnetoresistance. Since Ni is not a common material used in spintronic devices, and the AMR of ultrathin films can significantly deviate from their bulk behavior, a standard AMR characterization in the Ni-based systems would be useful to confirm that the ‘abnormal’ torque behavior is not an experimental artifact.

→ The reviewer questioned the possibility of AMR abnormality in Ni/Ta because V_s of ST-FMR is proportional to $\text{AMR} \times \text{SHA}$. We agree that the sign of V_s can be reversed not only by the abnormal sign of SHA but also by the abnormal sign of AMR. We performed AMR measurements for Ni(7)/Pt and Ni(7)/Ta to test the latter possibility. The results are shown in Fig. R1. Since the AMRs of both structures have the same sign, the sign of V_s is determined by the effective SHA that includes the orbital-torque contribution. Thus, we rule out the possibility that the abnormal sign of V_s in Ni/Ta is due to the reversed AMR sign of Ni. To explain this point, we added the following sentences to the manuscript: “As the sign of V_s depends not only on the sign of net spin Hall conductivity $\sigma_{net}^{FM/NM}$ but also on the sign of anisotropic magnetoresistance (AMR), we check the sign of AMR of Ni/Pt and Ni/Ta bilayers. We find that they are of the same sign so that the sign of DLT for Ni/Ta bilayer is abnormal.”

Figure R1. AMR measurement of (a) Ni(7) / Pt, (b) Ni(7) / Ta.

- The interfacial quality of Ni-based heterostructures should be adequately characterized and discussed to eliminate the possibility of new torques induced by strong interfacial alloying.

Depending on the outcome, this possibility should be discussed in the relevant section of the manuscript.

→ To exclude the possibility of the interfacial contributions between Ni and Ta layers, we showed the measurements with ultrathin Cu (down to 0.5 nm) intercalated samples. Note that a small amount of Cu acts as a surfactant that can suppress the interfacial reaction at the Ni/Ta interface [Thin Solid Films 484, 208-214 (2005), JAP 87, 5732 (2000)]. We found that the damping-like torque (DLT) still had the same abnormal sign as mentioned in the main manuscript (Fig. 4A, B) and the supplementary information (section S2). Thus, we conclude that the interfacial SOC effect or the interfacial alloying effect from the Ni/Ta interface is not the cause of the abnormal sign.

- The resonance peaks of the Ni-based structures are significantly broader with respect to the CoFeB-based structures. Why?

→ The FMR linewidth of Ni is broader than that of CoFeB because Ni has a stronger SOC than Fe and Co. In the FMR spectrum, the linewidth represents the magnetic damping of the bilayer system and can be regarded as a phenomenological parameter that describes how effectively the spin angular momentum of a magnetic system is dissipated into its environment (lattice). Among several contributions to the damping, it is known that the SOC of conduction electrons near the Fermi surface plays a major role in determining the intrinsic damping of magnetic metals. The breathing Fermi surface model of Kambersky [PRB 68,019901 (2003), PRL 99, 027204 (2008)] explains that the magnetic precessions change the energy of the electronic states via the SOC and induce breathing of the Fermi surface. This breathing pushes some occupied states above the Fermi level and some unoccupied states below the Fermi level, creating electron-hole pairs that will be relaxed through the lattice scattering. Hence, the amount of energy and angular momentum dissipated into the lattice depends on the SOC magnitude. The measured linewidths for Ni and CoFeB are consistent with our first-principle calculation (Section S1 and Fig. S2) and our experimental result (Section S5 and Fig. S13). Therefore, we believe that the reviewer's observation is consistent with the orbital torque scenario.

- Can the authors comment on the implication of these findings for other spintronic systems (magnetic insulator/NM, AFM/NM, etc.) and effects (SMR, SSE, etc.)?

→ As there has been no study on these issues yet, we below provide our speculations.

- Magnetic insulator/NM system: The (electrical) orbital torque (OT) requires the conversion from orbital currents to spin currents within a magnetic layer. As it is not possible for magnetic insulators, we expect no (electrical) OT in this system. However, if there is a process that converts orbital currents to magnon currents, (magnonic) OT may exist.
- AFM/NM system: If AFM is metallic, we expect that the (electrical) OT exists in this system because the mechanism of (electrical) OT requires SOC of magnetic layers, but not homogeneous exchange interaction.
- SMR, SSE, etc: We expect that orbital counterparts of these effects may exist because any orbital transport combined with SOC results in related spin phenomena.

Please note that all of the above statements are speculative and require further research in the future.

- The control experiment for the self-induced anomalous torque should be performed on the Ni layer alone and not on the Ni/Cu bilayer. Since Cu will have a significant shunting effect, it could be a good comparison to Ni/Pt but not to Ni/Ta, in which the authors observe the ‘abnormal’ behavior. An additional test could be Ni/Ti, provided that Ti is not a source of orbital angular momentum.

→ This is a very good point. Firstly, we estimate the magnitude of SOT generated by the ferromagnetic metal Ni alone and show that the magnitude cannot explain the abnormal sign of the DLT in the Ni/Ta bilayer. For this, we studied structures consisting of Ta(2)/MgO(2)/Ni(t_{Ni})/substrate. The thickness of Ni (t_{Ni}) is 7, 10 and 15 nm, and the substrate is a thermally oxidized Si substrate. Fabrication of the devices and ST-FMR measurements were performed in the same ways as described in the Materials and Methods section of the main text. Figure R2a shows a representative ST-FMR signal measured on a sample with $t_{Ni} = 7$ nm. We extract the DLT (FLT) conversion ratio ξ_{DL}^{Ni} (ξ_{FL}^{Ni}) of the Ni layer using the following equation. The voltage amplitude of the ST-FMR signal ($V_{S(A)}^{mix}$) is given by

$$V_{S(A)}^{mix} = -I_{rf}^2 \frac{\gamma}{4} \Delta R_{AMR,eff} \left[\frac{\sin 2\varphi \cos \varphi}{\Delta H \cdot 2\pi \cdot \left(\frac{df}{dH}\right)_{H_0}} \right] \left[\frac{\hbar}{2e M_s t_{FM}} \right] \left[\frac{1}{w \cdot t_{FM}} \right] \xi_{DL(FL)}^{Ni}, \quad (R1)$$

where $\varphi = \pi/4$ in our measurements and ΔH is a half-width at half maximum. The RF current through the device (I_{rf}) was estimated for a given RF power and frequency using a vector

network analyzer (VNA) and taking into account the loss factors. Effective AMR ($\Delta R_{AMR,eff}$) was also measured separately. Figure R2b shows the estimated ξ_{DL}^{Ni} and ξ_{FL}^{Ni} from the MgO/Ni/substrate layers. The averaged ξ_{DL}^{Ni} is 0.0010 ± 0.0008 , which is about 30 times smaller than the value from the MgO/Ni/Ta.

Figure R2. Estimation of SOTs from Ni itself. (a) Measured ST-FMR spectra from MgO/Ni(7)/substrate. (b) Estimated θ_{DL}^{Ni} and θ_{FL}^{Ni} .

Secondly, we estimate the upper bound of SHC of Ni from experiments for Ni/Cu bilayers with considering the shunting effect and assuming that the measured ξ_{DL} of Ni/Cu is entirely caused by the charge current flowing through the Ni layer (i.e., the SHE of Ni). Using a parallel resistor circuit of Ni and Cu, the upper bound of SHC of Ni is estimated to be about $+64$ (\hbar/e) $\Omega^{-1} \cdot \text{cm}^{-1}$, which is clearly smaller than the measured SHC of Ni/Ta (about $+146$ (\hbar/e) $\Omega^{-1} \cdot \text{cm}^{-1}$, Fig. S14b). We note that the measured SHC of Ni/Ta also includes the SHC of Ta, which is negative. The measured SHC of FeB/Ta is about -350 (\hbar/e) $\Omega^{-1} \cdot \text{cm}^{-1}$ (Fig. S14b), of which negative sign means that the negative SHC of Ta dominates over other contributions in FeB/Ta. Taking -350 (\hbar/e) $\Omega^{-1} \cdot \text{cm}^{-1}$ as the SHC of Ta, the combined contribution from orbital torque and anomalous torque in Ni/Ta is about $+500$ [= $+146 - (-350)$] (\hbar/e) $\Omega^{-1} \cdot \text{cm}^{-1}$, which is an order of magnitude larger than the estimated SHC (or anomalous torque contribution) of Ni [$+64$ (\hbar/e) $\Omega^{-1} \cdot \text{cm}^{-1}$].

The above results indicate the abnormal sign of the DLT for the Ni/Ta bilayer cannot be explained by the contribution from the Ni layer.

Thirdly, for the Ni/Ti system, we have calculated the OHC of Ti and found that it is about $3800 (\hbar/e) \Omega^{-1} \cdot \text{cm}^{-1}$ for the fcc structure, which is a stable crystal structure for thin films. Although this value is smaller than the OHC of Ta, it is still quite sizable. On the other hand, Zhu et al. (arXiv:2010.13137) reported recently that they failed to observe evidence of the OHE in Ti. So, we would say that the question of the OHE in Ti is rather controversial at the moment. For this reason, we did not carry out experiments for the Ni/Ti system.

Reviewers' Comments:

Reviewer #1:

Remarks to the Author:

After reading the detailed answers and new version of the manuscript, I believe it is now ready to publication in Nature Communications.

Reviewer #2:

Remarks to the Author:

The authors have satisfactorily addressed all of my concerns and the first reviewer's comments as far as I can tell. Therefore, I am now fully supportive of the publication of this manuscript in Nature Communications without further revision.